# Genetic Architecture and Genome-Wide Adaptive Signatures Underlying Stem Lenticel Traits in *Populus tomentosa*

**DOI:** 10.3390/ijms22179249

**Published:** 2021-08-26

**Authors:** Peng Li, Jiaxuan Zhou, Dan Wang, Lianzheng Li, Liang Xiao, Mingyang Quan, Wenjie Lu, Liangchen Yao, Yuanyuan Fang, Chenfei Lv, Fangyuan Song, Qingzhang Du, Deqiang Zhang

**Affiliations:** 1National Engineering Laboratory for Tree Breeding, College of Biological Sciences and Technology, Beijing Forestry University, No. 35, Qinghua East Road, Beijing 100083, China; lipeng@bjfu.edu.cn (P.L.); jiaxuanzhou@bjfu.edu.cn (J.Z.); 15684156282@163.com (D.W.); lzlee@bjfu.edu.cn (L.L.); xiaoliang0622@126.com (L.X.); Mingyangquan@bjfu.edu.cn (M.Q.); wenjielu@bjfu.edu.cn (W.L.); liangchenyao@bjfu.edu.cn (L.Y.); yuanyuanfang@bjfu.edu.cn (Y.F.); chenfeilv@bjfu.edu.cn (C.L.); fangyuansong@bjfu.edu.cn (F.S.); Qingzhangdu@bjfu.edu.cn (Q.D.); 2Key Laboratory of Genetics and Breeding in Forest Trees and Ornamental Plants, Ministry of Education, College of Biological Sciences and Technology, Beijing Forestry University, No. 35, Qinghua East Road, Beijing 100083, China

**Keywords:** GWAS, lenticel, local adaption, *PtoNAC083*, *Populus*, selective signatures

## Abstract

The stem lenticel is a highly specialized tissue of woody plants that has evolved to balance stem water retention and gas exchange as an adaptation to local environments. In this study, we applied genome-wide association studies and selective sweeping analysis to characterize the genetic architecture and genome-wide adaptive signatures underlying stem lenticel traits among 303 unrelated accessions of *P. tomentosa*, which has significant phenotypic and genetic variations according to climate region across its natural distribution. In total, we detected 108 significant single-nucleotide polymorphisms, annotated to 88 candidate genes for lenticel, of which 9 causative genes showed significantly different selection signatures among climate regions. Furthermore, *PtoNAC083* and *PtoMYB46* showed significant association signals and abiotic stress response, so we overexpressed these two genes in *Arabidopsis thaliana* and found that the number of stem cells in all three overexpression lines was significantly reduced by *PtoNAC083* overexpression but slightly increased by *PtoMYB46* overexpression, suggesting that both genes are involved in cell division and expansion during lenticel formation. The findings of this study demonstrate the successful application of an integrated strategy for dissecting the genetic basis and landscape genetics of complex adaptive traits, which will facilitate the molecular design of tree ideotypes that may adapt to future climate and environmental changes.

## 1. Introduction

Many long-lived perennial trees show local adaptation responses to multiple abiotic or biotic factors across large geographic ranges, spanning widely variable spatial environments [1]. A series of stable adaptive phenotypes have formed during the long course of their evolution, including bud break, flowering, leaf morphology, and bark composition [2,3,4]. The divergence of selection pressures among local environments shapes stable local ecotypes with unique adaptive phenotypes, producing remarkable selection signatures in the genomes of present-day ecotypes [5]. For example, differences in winter temperatures among a range of latitudes has produced significant differences in the bud break of *Populus trichocarpa* and altered the frequency of its genomic variations [2,6]. A deeper understanding of the genetic basis of local adaptation in tree species and the determining factors will offer insights into tree genetic improvement and ecosystem management in natural populations [5,7].

Morphological adaptations, such as leaf shape and bark texture, are among the most tangible examples of tree adaptations [8,9]. The stem lenticel is a highly specialized tissue of woody plants that has evolved to balance stem water retention and gas exchange as an adaptation to local environments, and it shows broad environmental diversity among tree species. Most trees contain lenticels within their rough bark, making them difficult to observe in genera such as *Ulmus* and *Quercus*, whereas lenticels are clear and observable in species such as *Prunus persica* and *Populus tomentosa*. Anatomically, the lenticel is a type of filling tissue that can be produced annually; it arises from phellogen, which is continuous with the surrounding corky periderm, but more active (Appendix A) [10].

The fundamental role of lenticels is to support stem water evaporation and the exchange of gases such as carbon dioxide and oxygen, and to regulate light penetration into the stem [11,12]. Most stem moisture is lost through lenticel tissue, although it covers only 3% of the stem surface of species such as *Betula pendula* [13]. The loosely arranged lenticel tissue can also act as a channel for invasion by pests and diseases [14]; for example, *Pityogenes chalcographus* invades *Picea abies*, and *Pseudomonas syringae* invades *Aesculus hippocastanum* through the lenticel [15,16]. The structural features and biological functions of the lenticel and periderm have been well characterized in trees [17,18], but the genetic architecture and adaptive mechanisms remain poorly understood. 

Similar to other tree quantitative traits produced by secondary growth, lenticels show wide phenotypic variation across ecotypes; therefore, advanced genetics approaches such as genome-wide association studies (GWAS) are suitable for examining the genetic basis of the complex traits of lenticels [2,5,19,20]. The genomic selection signatures underlying these adaptive traits can be identified using selective sweeps [21], which allow us to evaluate the role of local adaptation in shaping the genetic variation of entire species [22,23,24].

The genus *Populus* comprises 25–35 commercially and ecologically important species of deciduous woody plants native to large geographic areas of the Northern Hemisphere [25]. Lenticel morphology is relatively uniform among *Populus* species, but some species display rich phenotypic variation, including *P. tomentosa*, which typically shows diamond-shaped lenticels on its smooth bark but diverse stem lenticel sizes and numbers across its broad geographic distribution in China (Appendix A). This diversity leads to significant environmental adaptability and provides an opportunity for investigating the genetic basis of adaptive traits. 

In the present study, we applied GWAS to examine the genetic basis (additive, dominant, and epistatic effects) of variation in lenticel morphology in a natural population of 303 unrelated *P. tomentosa* accessions. We also performed selective sweeps to examine the geographic distribution of causative variation underlying adaptive lenticel traits. We characterized the functions of *PtoNAC083* and *PtoMYB46* from *P. tomentosa* by overexpressing them in *A. thaliana*. The approaches used in this study will improve our understanding of the genetic basis underlying adaptive traits for the molecular design of improved varieties in different climate and geographic regions.

## 2. Results

### 2.1. Phenotypic Variation in the Lenticel Traits of the Natural P. tomentosa Population

To assess phenotypic variation in the natural population, we quantified lenticels in terms of single lenticel area (LA, cm^2^), lenticel number (LN), and ratio of the total lenticel area to the total area of a rectangle (RA). All three lenticel traits showed considerable phenotypic diversity in the natural population, with coefficients of variation of 26.2–39.6% (Appendix A). For example, LN ranged from 5.25 to 25 within the sampling area, which demonstrates its potential for population genetics and improvement studies (Appendix A). Lenticel traits also showed high clonal repeatability, with values ranging from 0.460 (LA: *p* = 4.86 × 10^–40^) to 0.549 (LN: *p* = 4.67 × 10^–56^) (Appendix A). Correlation analysis of the three traits showed that LA and LN were positively correlated with RA (LA: *r* = 0.775, *p* = 2.81 × 10^–55^; LN: *r* = 0.481, *p* = 5.29 × 10^–17^), whereas LA and LN had a weak negative correlation (*r* = −0.149, *p* = 0.014). LA had a weak positive correlation with diameter at breast height (*r* = 0.151, *p* = 0.013; Appendix A), which suggests that pleiotropic loci may control these correlated traits.

Next, we compared phenotypic variation in the three lenticel traits between subpopulation pairs from three corresponding climate regions, which we presume to have adapted to their different environments (Appendix A). We detected significant differences in the lenticel phenotypes across the three subpopulations (Figure 1A–C and Appendix A). For example, the mean LA was higher in the Northeast (NE) than in the Northwest (NW) and South (S) subpopulations (Figure 1B,C and Appendix A), whereas the highest LN value was observed in the S, followed by NE and NW, subpopulations (Appendix A). Similarly, the S and NE climate zones tended to have higher RA values than that of the NW zone (Appendix A), which is consistent with differences in rainfall, mean annual temperature, and relative humidity among the three climate zones (Appendix A). 

### 2.2. Additive and Dominant Effects of Natural Variants Underlying Lenticel Traits Determined by GWAS

We conducted GWAS under the condition of taking population structure and genetic kinship as covariates to reveal additive and dominant effects among 9,401,992 high-quality SNPs and three lenticel traits in 303 *Populus tomentosa* accessions. 112 significant associations (*p* < 5.318 × 10^–8^) were identified among 108 SNPs and three lenticel traits, and each SNP explained 5.3–17.9% of the phenotypic variation (Appendix A). Next, we annotated 88 genes in the 5-kb regions upstream and downstream of 108 significant SNPs using the *P. tomentosa* reference genome (Appendix A). GO analysis revealed that the 88 genes extensively participated in the second cell biogenesis and transcription regulation (Appendix A).

Considering the genetic effects of the association loci, 88 and 80 associations showed significant additive and dominant effects, respectively, and 58 exhibited both additive and dominant effects (Appendix A). Five association loci annotated with 5 pleiotropic genes, including 2 unknown protein-coding genes (Appendix A), shared different pleiotropic effects between LA and RA traits, indicating their critical roles in lenticel formation. Among these pleiotropic SNPs, they have either an additive or dominant effect for different lenticel traits (Appendix A). For example, SNP Chr3_4043365 exhibited an additive effect for LA but additive and dominant effect for RA. We further identified that SNP Chr14_5078112 exhibited joint effects for LA and RA, different additive effects for LA (−0.16) and RA (0.05), and dominant effects for LA (−0.11) and RA (0.07). These findings suggest complex genetic effects of polygenic loci during the formation of lenticel variation.

### 2.3. Pairwise Epistatic Effects of Natural Variants Underlying Lenticel Traits

To decipher the potential gene networks involved in lenticel traits, we focused on pairwise epistatic interactions for each trait among the 108 associated loci. In total, 15 significant epistatic pairs were detected for the three lenticel traits at a threshold of *p* < 0.001 (Appendix A), indicating that epistasis provides important information about the genetic interaction networks underlying lenticel trait variation. For example, we found that a major associated SNP, Chr3_4043365 (Figure 2A,B), which is located in the promoter region of *Ptom.003G.00501* (*PtoNAC083*), significantly interacted with five unique SNPs for the three lenticel traits (Appendix A). We also found that allele interactions of different genotype combinations were significantly different from those of a single locus. For example, SNP Chr3_4043365 (T/C) had significant epistatic interactions with SNP Chr9_3551378 (G/A), which belongs to *Ptom.009G.00386* (*PtoMYB46*), for LA. The genotype combination of T (Chr3_4043365) and G (Chr9_3551378) showed the minimum LA, whereas that of C (Chr3_4043365) and A (Chr9_3551378) showed the maximum LA (Figure 2C). 

### 2.4. Differences in Allele Frequency of Associated Genes among P. tomentosa Subpopulations 

Based on the significant phenotypic differences across the three subpopulations from different climatic regions (Figure 1), we computationally examined genomic signatures that appeared to be affected by adaptive selection of local environment by using a 5 kb non-overlapping sliding window across the whole genome among NE, NW, and S climatic regions. Combining the adaptive selection signatures and GWAS signal results, we identified 9 representative genes, which harbored SNPs that were significantly associated with lenticel traits, and the gene regions overlapped the genomic regions of adaptive selection among the climate regions of *P. tomentosa* (Table 1 and Figure 3A–C). These 9 genes were significantly associated with the lenticel traits and were significantly selected by the local environment, suggesting that these genes have important effects on the lenticel and thus the adaptation of local environment.

Genes associated with LA exhibited a significant difference in frequency between the NE subpopulation and the NW or S subpopulation (Figure 3D). For example, *TREHALASE*
*1* (*Ptom.003G.00730*), which had a significant SNP (G/T) for LA (*p* = 1.288 × 10^–8^), showed obvious differences among the subpopulations, in that the mutant allele (T) was dominant in the NE subpopulation but was missing from the NW and S subpopulations (Figure 3D–E). We also discovered four associated SNPs for the LN trait with significant differences in allele frequency among the three climate regions (Figure 3F). For example, a synonymous variant, SNP Chr10_7228249 (A/T) in intron of *Ptom.010G.01098*, encoding Cytochrome P450 superfamily protein, showed obvious differences among the subpopulations, whereas the mutant allele (T) was decreased in frequency in the low-latitude S subpopulation compared with the NE and NW subpopulations. Similar differences were observed in Chr17_4717361, Chr2_18582313, and Chr6_6559040, which are presumably mutant alleles with higher frequencies in northern China (Figure 3F; Table 1).

The specific environment of each climate region is likely to shape phenotypic variation by influencing the genotype frequency of causative genes. Indeed, the allele frequency patterns of the associated loci were consistent with lenticel phenotypic differentiation among subpopulations (Figure 3D–G). For example, this phenomenon was observed for the most likely causal gene, *Ptom.017G.00466* (*KTN1*; *KATANIN 1*), which influences multidimensional cell growth and plant cell wall biogenesis in *A. thaliana* [26]. A SNP (Chr17_4717361; A/G) within *KTN1* was strongly associated with LN (*p* = 7.5907 × 10^–9^) and was located in the selective signal peak with strong π and F_ST_ outliers (Figure 3B,C). Thus, SNP Chr17_4717361 showed obvious frequency differences among the subpopulations and corresponding region-specific additive effects, in which accessions in the NW subpopulation with the mutant homozygous allele (G) had a smaller LN compared with those with the major homozygous allele (A), which was distributed mainly in the S subpopulation, and accessions in the NE subpopulation with heterozygous genotypes (A/G) had intermediate phenotype values (Figure 3E).

### 2.5. Allelic Interpretation of the Potential Causal Genes Underlying Lenticel Variation

Our GWAS revealed the most significant variants involved in lenticel variation, including the top SNPs (with the most significant *p* values) or high-LD haplotypes of multiple top-ranked associated SNPs underlying lenticel traits in *Populus*, which may represent causative signals for each trait (Table 2). For example, we detected a non-coding SNP (Chr7_10376607, A/C) within the fifth intron of *PtoTraB2* (*Ptom.007G.00998*), which was strongly associated with LN (*p* = 1.15 × 10^–10^) and is likely to be a causative gene within the associated signal (Figure 4A–C). Accordingly, *TraB1* (*Ptom.007G.00997*), another member of the TraB family in *P. tomentosa*, was found to be located 634 bp up.

Stream of *TraB2*. We compared the expression patterns of both genes across various tissues of the *P. tomentosa* clone (Figure 4D). Both members showed different tissue-specific expression patterns; *PtoTraB2* was highly expressed in bark and *PtoTraB1* in mature xylem (Figure 4D). qRT-PCR showed significantly higher *PtoTraB2* expression in more lenticel numbers than in the general *P. tomentosa* population, revealing a function of *PtoTraB2* in regulating LN in *P. tomentosa* (Figure 4E). Next, we further analyzed *PtoTraB2* as a potential variant underlying LN. A portion of the 11,560-bp genomic region, spanning from the promoter to the 3′-untranslation regions (UTR) of *PtoTraB1* and *PtoTraB2*, was subjected to candidate gene-based association fine mapping. A newly identified InDel (Chr7_10376577, CTTATTTCT/C) had the most significant association with the LN (*p* = 1.67 × 10^–8^), which was in high LD with the SNP Chr7_10376607 (r^2^ = 0.96), identified in GWAS (Figure 4F–H). The double minor haplotype homozygous (C-C) accessions had greater LN values and higher gene expression levels of *PtoTraB2* than those of the double WT haplotype homozygous (CTTATTCT-A) accessions. The common haplotype heterozygote (CTTATTCT/C-A/C) had moderate LN values compared with the double homozygote groups, which revealed an additive effect of the haplotype on lenticel variation (Figure 4E).

### 2.6. A NAC-Type Transcription Factor (PtoNAC83) Is an Important Genetic Regulator Contributing to Lenticel Variation

To identify the possible causative variants within the representative *PtoNAC083* locus for lenticel variation, the association analysis was carried out with the *Populus tomentosa* population of 303 10-year-old accessions that were planted in a common garden in Guan Xian County, Shandong Province, China. In total, 46 InDels and 54 SNPs located in the promoter and coding region of *PtoNAC083*, respectively, and two lenticel traits (LA and RA) were examined statistically. Four significant associations were detected in two traits, representing a known SNP (Chr3_4043365) and a new InDel (Chr3_4043376) in *PtoNAC083* (Figure 2C,D). The InDel Chr3_4043376, AGG/A, located in the promoter region of *PtoNAC083*, was the lead variant for LA (*p* = 4.06 × 10^–8^) and RA (*p* = 5.16 × 10^–8^). qRT-PCR showed significantly higher *PtoNAC83* expression in accessions with common genotype (AGG) than in the accessions with minor genotype (A), revealing that a causal variants InDel Chr3_4043376 is most likely a causal, or is in high LD with the causal variants responsible for the natural of LA, as well as *PtoNAC083*expression (Figure 2C,D,F). In addition, SNP Chr3_4043365 had significant epistatic interactions with SNP Chr9_3551378 (G/A), which belongs to *PtoMYB46*, for LA (Figure 2E; Appendix A). These two genes also showed opposite expression patterns in lenticels with different alleles (Figure 2F), which indicates that *PtoNAC083* and *PtoMYB46* play opposite roles in lenticel phenotypic variation. Annual rainfall and temperature are the main environment factors among three climate regions [10,27]. We further compared the expression abundance of *PtoNAC083* and *PtoMYB46* under drought and normal conditions in a population constituted with 100 accessions of *P. tomentosa* [28]. The expression of *PtoNAC083* in drought environment is higher than that in normal environment, and the expression of *PtoMYB46* in drought environment is lower than that in normal environment (Appendix A). NW is located in arid and semiarid areas, which experience little precipitation and a wider annual temperature range than the NE. These results indicate that *PtoNAC083* and *PtoMYB46* show consistent expression patterns in natural population (constituted with 303 accessions) and drought stress population (constituted with 100 accessions).

Finally, we heterologously overexpressed *PtoNAC083* and *PtoMYB46* in *A. thaliana* and measured cell traits in three independent homozygous transformed lines (Figure 5). Although *A. thaliana* cannot produce lenticels, this species is associated with the proliferation and differentiation of phellogen (Appendix A); therefore, we used cell number and size to evaluate the roles of the key genes. Stems of the *PtoNAC083-*overexpressing plants were markedly thinner (56.16–78.23%), whereas those of *Pto**MYB46-*overexpressing plants were thicker (119.40–155.71%) than those of WT plants (Figure 5A). We further examined cell morphology and numbers and found decreases of 26.60% and 38.22%, respectively, in *PtoNAC083-*overexpressing plants, which suggests that these plants developed thinner stems with fewer and smaller cells. In *Pto**MYB46-*overexpressing plants, the cell number and size were increased by 9.04% and 29.67% (Figure 5C–E), respectively, which suggests that *PtoNAC083* restricts LA and RA by limiting cell number and area, and that *Pto**MYB46* promotes lenticel production by increasing cell number and area, although this result was not significant. In addition, there is no significant difference in stomatal morphological (stomatal length, stomatal width, and area) and net photosynthetic rate of overexpressing plants (Figure 5F–G,I). However, the *PtoNAC083-*overexpressing and *Pto**MYB46-*overexpressing plants showed slower transpiration rate and higher water use efficiency than WT plants (Figure 5J,K).

## 3. Discussion

Forest trees have evolved long-term adaptive traits to adapt to land ecosystems. Lenticel tissue is characterized by small pores on stem surfaces that are composed of filling tissue and phellogen; these pores provide channels for the direct exchange of gases between internal tissues and the atmosphere [29,30]. Using GWAS and selective sweep analysis, we detected the genetic architecture and adaptive selection signals of lenticel morphology and identified the causative loci underlying its intra- and interpopulation variations in natural populations of *P. tomentosa*. The results of this study improve our understanding of the genetic basis and genomic differentiation of complex plant traits that are difficult to quantify using traditional research methods.

### 3.1. GWAS Permits Comprehensive Understanding of the Genetic Basis of Lenticel in Populus

GWAS is a powerful tool that can be applied to determine the causal genes controlling quantitative trait variation in trees by combining phenotype and genotype data in population genetics studies [31,32]. In the present study, we detected 108 SNPs representing 88 genes associated with three lenticel traits (Appendix A), representing a largely polygenic genetic basis for lenticel in a natural population of *P. tomentosa*. Functional annotation of these genes showed that they belong to different functional categories, including regulation of cell growth, cell wall biogenesis, and stimulus response (Appendix A). Similar results were obtained in a genetic study of the secondary growth processes of woody plants [33], suggesting that lenticel formation and development are controlled by multiple factors affecting cell division and expansion, in a process similar to wood formation [34]. Among these genes, three were previously reported to participate in bark formation in other poplar species [8,35,36]. Considering that bark and lenticel develop from phellogen, these three genes may reflect aspects of the commonalities between the two tissues. For example, we identified an association of *Ptom.016G.00430*, a homolog gene encoding a NIT4 enzyme (AT5G22300), with LA (Appendix A). Previous studies have demonstrated that *NIT4* plays a key role in cyanide detoxification during ethylene biosynthesis in *Arabidopsis* [37]. Additional evidence has shown that *Ptom.016G.00430* is associated with bark texture and may be related to the characteristics of bark and lenticel, which are both composed of dead differentiated cells and are produced in annual strata [8,10].

We further categorized the genetic basis underlying lenticel variation into the additive and dominant contributions of alleles to provide a theoretical foundation for the construction of genetic control networks for complex traits. Pleiotropic loci have diverse additive/dominant effects for different lenticel traits. For example, Chr3_4043365, a pleiotropic locus for LA and RA, showed negative additive/dominant effects (−0.16/−0.11) for LA but a positive additive/dominant effect for RA (0.05/0.07) (Appendix A). Thus, these loci affect lenticel traits via a combination of additive and non-additive alleles, providing a foundation for genetic improvements in *P. tomentosa*, such as landscape tree species with smooth and attractive bark via molecular design.

Epistatic interactions generally define the non-additive interactions between two associated loci [38]. Our epistasis analysis identified 15 significant epistatic pairwise interactions among three lenticel traits, including eight loci with additive/dominant effects (Appendix A), which indicates that epistasis could provide important information about the interactive genetic networks underlying lenticel. For example, a top SNP, located in the promoter region of *PtoNAC083*, had significant epistatic interactions with Chr9_3551378, which is found in *PtoMYB46*. *PtoNAC083* and *PtoMYB46* also showed opposite tissue-specific expression patterns, indicating a potential interaction between the two genes, which is consistent with prior studies of *A. thaliana* reporting that the homologous *NAC083* (*VND-INTERACTING2*, *VNI2*) inhibits transcriptional activation of *MYB46* in the regulatory network of xylem vessel formation [39]. Epistasis is important for further exploration of the mechanisms of plant responses to environment selection [40]. Interestingly, Chr3_4043365, a likely causal locus for lenticel variation, showed significant epistatic interactions with two SNPs that are close to selective sweep signals, suggesting that lenticel phenotypic variation responds to adaptive selection mainly via gene regulation by a hub gene rather than direct effects [41,42]; these findings may improve our understanding of the roles of epistasis in environmental adaptation in tree species [5,22]. The four components of the epistatic effect (additive × additive, additive × dominant, dominant × additive, and dominant × dominant) are related to allele frequency [43], and the allele distributions of these two adaptability-related loci showed significant differentiation among subpopulations (Figure 3E). These results may indicate different epistatic patterns in individual forest trees to adapt to the local environment. Therefore, further epistasis studies on different subpopulations may expand our understanding of the ability of forest trees to adapt to local ecosystems.

### 3.2. Local Adaptation of Candidate Genes with Overlapping Selection Signals and Their Associated Loci

The lenticel is a morphological adaptive trait of forest trees, with geographically structured phenotypic variation that is significantly affected by both biotic and abiotic stresses [5,16,44]. In this study, we used natural populations of *P. tomentosa* spanning wide climate gradients with significant spatial and geographic structures [22,45]; this system offers an ideal opportunity to examine the environmental forces shaping lenticel phenotypic variation. We observed lenticel traits that underwent selection pressure responsible for local adaptation among different climate regions of *P. tomentosa* (Figure 1 and Appendix A). The S climate region has a low latitude and subtropical monsoon climate, in which accessions receive abundant water and heat but frequently experience flooding and high-temperature stress [45]. Local accessions typically exhibit a smaller LA and greater LN and RA, which may allow rapid gas exchange to meet growth demands and resist flooding and disease stresses [15,46,47]. In contrast, the NW subpopulation is located in arid and semiarid areas which experience little precipitation and a wide annual temperature range. Accessions in this area with low LA, LN, and RA values can prevent water loss during drought [11]. Thus, lenticel formation may be affected by environmental conditions, leading to local variations in characteristics such as stem growth [22,48]. Even though association genetics methods provide estimates of the genetic architecture of complex traits, they cannot inform the relationships between individual genomic regions and adaptive traits. Selective sweep is a powerful tool that can be used to identify genomic regions related to local adaptation signals in trees, which would provide further information about the mechanisms by which subpopulations adapt to their local environments [48].

In this study, we identified 9 genes with overlapping GWAS and selective sweep signals (Table 1 and Figure 3), suggesting that these causal genes underlying lenticel formation are selected by the local environment. The functional categories of these genes contain multiple annotations related to cell signal transduction, cell growth, and stress response (Appendix A). Among these genes is *Ptom.010G.01894*, and its homologous gene, *AT3G25400*, affects sugar transport and/or interconversion of applied sugars in *Arabidopsis* [49]. The homologous gene of *Ptom.006G.00836* in *Arabidopsis* is *AT3G53900*, which affects growth by regulating pyrimidine catabolism [50]. The homologous gene of *Ptom.003G.00730* is *AT4G24040* (*TRE1*), which is involved in trehalose catabolism and plays an important role in the plant responses to cold and salinity stresses [51]. Based on our functional categorization, these 10 genes may involve lenticel adaptation to the environment in two ways. First, compared with the surface of bark, which is covered with waxy cutin and suberin, that of the lenticel consists only of cell wall outgrowth, which is insufficient for protection against diverse environmental pressures [10,52]. Therefore, lenticel cells have evolved different adaptive mechanisms to enhance their adaptability to the local environment; for example, some genes may affect trehalose biosynthesis to enhance cell adaptation in cold and high-salinity environments [51]. Second, poplars adapt to the local environment via genetic regulation of the number and/or morphology of lenticel cells to achieve gas exchange in the stems [29].

We further investigated population differences in the allele frequency and geographic distribution of gene loci with major effects among candidate genes, with overlapping GWAS and selective sweep signals (Figure 3D–G). The mutant allele of the significant SNPs associated with LN was distributed mainly in the NW region, followed by the NE region, and the mutant allele of the SNPs associated with LA was distributed mainly in the NW region, followed by the NE or S region (Figure 3D). These findings provide direct molecular evidence of the mechanism underlying the local adaptation of lenticel traits. For example, the mutant allele associated with Chr10_7228249 (A/T), a SNP associated with LN, was increased in frequency in the S subpopulation compared with the NE and NW subpopulations, perhaps due to the significantly colder and drier environment. The regional differentiation patterns of these loci/genes are partly consistent with the geographic distribution of lenticel traits (Appendix A and Figure 3; Appendix A), which suggests that the geographic environment had an extensive impact on lenticel performance during the adaptive evolution of *P. tomentosa*. Thus, the geographic distribution of favorable alleles sheds light on the potential for tree improvement in future breeding programs.

### 3.3. PtoNAC083 Is a Causal Gene Underlying Lenticel Formation in P. tomentosa

Unlike the adaptation-related genes with overlapping GWAS and selective sweep signals, we identified potential causative signals underlying *Populus* lenticel traits using a single GWAS. These genes may be involved in multiple biological processes, from secondary cell wall biogenesis and stress response to growth regulation and environmental adaptation (Appendix A). We further examined locus *PtoNAC083* as a representative example and found that in the promoter region of *PtoNAC083*, the top-ranked variant SNP Chr3_4043365 (T/C) was simultaneously associated with LA and RA (Figure 2A; Appendix A), which is consistent with the higher expression of *PtoNAC08* in bark (Appendix A).

NAC proteins possess a conserved NAC domain that participates in various secondary growth and stress resistance processes in *Populus* [17,53,54,55]. *NAC083* modifies cambial growth and cell differentiation by signaling and transcriptional reprogramming via ethylene [56]. *PtoNAC083* is homologous to *VNI2*, a key transcriptional repressor of xylem vessel development in *A. thaliana*. *VNI2* overexpression represses the normal differentiation of xylem vessels [39], which is useful for exploring novel functions of *NAC* in secondary growth. Our findings revealed that locus *PtoMYB46* significantly interacted with *PtoNAC083* to affect LA, which is consistent with the opposite tissue-specific expression patterns of the genes in *P. tomentosa* (Figure 2F; Appendix A). These epistatic findings are supported by reports that *VNI2* represses *MYB46* expression during xylem vessel formation in *A. thaliana* [39,57]. *PtoNAC083* and *VNI2* share a typically conserved N-terminal DNA domain and 59% sequence similarity at the full protein level. To further explore the function of *PtoNAC083*, we generated transgenic *A. thaliana* overexpressing *Pto**NAC083* and *Pto**MYB46*. Histochemical staining of stem cross-sections of the transgenic *Pto**NAC083* lines showed decreased cell number and sizes, suggesting that *Pto**NAC083* may influence lenticel variation by cell number and sizes. Consistently, overexpression of *Pto**MYB46* resulted in increased, albeit not significantly, cell number and size (Figure 5E–G). Overexpression of *Pto**NAC083* and *Pto**MYB46* showed opposite phenotypes, which supports the significant epistatic interaction between SNPs Chr3_4043365 and Chr9_3551378; in turn, the combination of these two loci has significant phenotypic effects. Thus, the phenotype of an individual depends on the specific combination of alleles at two loci, which may inform the functional interaction between these two genes [38]. However, the present study explored the relationship between these genes only at the expression level, without further examination of the molecular biological functions of *Pto**NAC083* and *Pto**MYB46* in poplar. The epistatic genetic effects of Chr3_4043365 and Chr9_3551378 and the opposite phenotypes induced by *Pto**NAC083* and *Pto**MYB46* overexpression further demonstrate the important role of *Pto**NAC083* in lenticel formation, as well as the importance of epistasis analysis for systematic genetic analysis of quantitative traits. Because stem lenticels are produced during secondary phellogen growth, which is a multi-year accumulative process in *P. tomentosa*, it is difficult to verify the key genes involved in lenticel formation via gene transfer. In this study, we selectively verified the functions of *PtoNAC083* and *PtoMYB46* in *Arabidopsis* using a gene transfer approach, but the direct function of *NAC083* in *P. tomentosa* requires further investigation. Lenticel formation usually begins beneath stomatal complexes during primary growth preceding the development of the first periderm [10], but there are no significant differences in the stomatal morphology of *PtoNAC083-*overexpressing and *Pto**MYB46-*overexpressing plants compared with WT (Figure 5F,G), which may be related to the formation of lenticels in the secondary growth process [10]. In addition, *PtoNAC083-*overexpressing and *Pto**MYB46-*overexpressing plants showed slower transpiration rate and higher water use efficiency than WT plants (Figure 5J,K), which may indicate that these two genes may be involved in stress response. Previous studies of homologs of *PtoNAC083* and *PtoMYB46* have shown that they play an important role in plant response to stress. For example, *ANAC083* in *Arabidopsis* is induced by salt stress [58]; *ZxNAC083* in *Zygophyllum xanthoxylum* conferred tolerance to salt and drought stress when constitutively overexpressed in *Arabidopsis* plants [59]; *MdMYB46* in apple could coordinate stress signal transduction pathways to cooperate with the formation of secondary walls to enhance the stress tolerance [60]. Considering that the lenticel traits that underwent selection pressure responsible for local adaptation among different climate regions of *P. tomentosa* (Figure 1 and Appendix A), *PtoNAC083* and *PtoMYB46*, responded to stress, may provide further support for these two genes involved in lenticel traits.

In summary, this study is the first to identify the genetic architecture and adaptive mechanisms of lenticel variation. The integration of multiple strategies, including GWAS (additive, dominant, and epistatic effects), selective sweep, gene expression profiling, and *Arabidopsis* overexpression allowed us to identify candidate genetic factors underlying lenticel morphology and may also provide insights into the development of similar quantitative traits in perennial trees [22]. In this field, the joint application of a high-throughput phenome platform and multi-omics data may enable further exploration of the genetic basis of complex quantitative traits at the phenotype and biological system levels. Further studies of the interactions between trees and microbial communities in the lenticel may provide important information about tree adaptation to the local environment.

## 4. Materials and Methods

### 4.1. Association Population and Phenotyping

The association population used in this study consisted of 303 10-year-old accessions that were planted in a common garden in Guan Xian County, Shandong Province, China (36°23′ N, 115°47′ E). In 2009, the association population was asexually propagated via root segments and planted in a randomized complete block design with three replications (blocks). The association population was randomly selected from a collection of 1047 natural *P*. *tomentosa* accessions [61], representing almost the entire species range (30–40° N, 105–125° E), which has been previously divided into southern (S, *n* = 102), northwestern (NW, *n* = 93), and northeastern (NE, *n* = 108) climate regions of China [45].

A digital camera (Canon EOS 800D) was used to capture images of lenticels at a tree height of 1.35 m for 303 accessions in July 2019. To control environmental effects, phenotypes of all 303 accessions (each accession contains three replications) were measured from the directions (south, west, north, and east), and the mean value of 12 data (3 replicates × 4 directions) represented the final phenotypes of each accession. A rectangle with a horizontal width of 5 cm and vertical length of 10 cm was used as the standard sampling area (only lenticels within the range are counted). Images were analyzed to estimate the single lenticel area (LA, cm^2^), lenticel number (LN), and ratio of the total LA to the total area of a rectangle (RA) within the sampling area (5 × 10 cm) using ImageJ software [62]. We also measured the tree diameter at breast height (cm). Clonal repeatability was estimated using the rptR package ver. 0.9.22 in R ver. 3.5.3 software. Descriptive analysis, multiple comparisons, and correlation analysis of the phenotype data were conducted using one-way analysis of variance, followed by the least significant difference and Pearson correlation tests, using SPSS ver. 19.0 software (SPSS Inc., Chicago, IL, USA).

### 4.2. Resequencing and Single Nucleotide Polymorphism (SNP) Calling

Preparation of the data used in this study has been described previously [63]. First, total genomic DNA was extracted from fresh leaves of 303 unrelated accessions using the DNeasy Plant Mini kit (Qiagen, Shanghai, China) and was used to construct resequencing libraries according to the manufacturer’s sample preparation instructions (Illumina, Tianjing, China). Then, we re-sequenced 303 genotypes at a depth of > 15× (raw data) using the Illumina GA II platform, following the manufacturer’s recommendations, to generate raw reads (100-bp or 150-bp paired-end reads). For quality control, we removed raw reads shorter than the length threshold (≤ 50% of nucleotides with a quality score < Q20). Short reads were then aligned to the *P. tomentosa* genome using the BWA-MEM algorithm in the BWA software package [64]. SNP calling was performed using the Genome Analysis Toolkit ver. 4.0 and the S_AMTOOLS_ ver. 1.3.1 software [65,66] In this process, low-quality SNPs with a QualByDepth < 2.0, Fisher Strand > 60.0, StrandOddsRatio > 3.0, quality < 20, depth of coverage < 8, and RMSMappingQuality < 40.0 were filtered out. Insertion/deletion (InDel) calling was performed using a QualByDepth < 2.0, quality < 20, Fisher Strand > 200.0, StrandOddsRatio > 10.0.

### 4.3. Population Genetics Analysis

We filtered the common SNPs using two criteria: minor allele frequency (MAF) < 0.05 and r^2^ (measure of linkage disequilibrium [LD]) < 0.2. The resulting 410,171 SNPs were used for kinship and population estimates. Varying levels of K (1–13) in our association population were calculated using the Bayesian clustering program fastStructure, and an optimal K value of 3 was selected using the chooseK.py script. Therefore, we used the K = 3 cluster result for the Q matrix as a covariant in our GWAS. We calculated genome-wide LD across the *P. tomentosa* genome using all SNPs with a MAF > 0.05 using the command “plink—vcf vcf_file—allow-no-sex—maf 0.05—r^2^—ld-window 5000—ld-window-r^2^ 0.2—out out_file” in the PLINK software [67].

### 4.4. GWAS Test and Candidate Gene Association Analyses

We performed mixed linear model association analysis of three lenticel traits using the set of SNPs remaining after the removal of low-frequency SNPs (MAF < 5%) and LD-correlated SNPs (r^2^ > 0.7) in the TASSEL ver. 5.0 software [68]. We use PLINK to generate LD-correlated SNPs (r^2^ > 0.7) with the following parameters: indep-pairwise 50 5 0.7. The lenticel traits (LA, LN, and RA) used in GWAS were transformed by Z-score normalization. The population structure (Q) was evaluated using the STRUCTURE ver. 2.3.4 software [69], and pairwise kinship coefficients (K) were assessed using the SPAGeDi ver. 1.3 software [70]. The significance thresholds of the GWAS were determined using the modified Bonferroni correction [71], which suggested a threshold of *p* ≤ 5.318 × 10^–8^ (*p* = 0.5/*n*, where *n* = total SNPs used in association studies [9,401,992 SNPs]). Then, we defined 5-kb windows centered on each significant SNP. All gene models of *P. tomentosa* that overlapped a window were selected as candidate genes. When a window overlapped within more than one gene, we annotated the SNPs with multiple genes. LD was determined using the HAPLOVIEW ver. 4.2 software [72]. Manhattan plots were generated in R using the ‘qqman’ package [73].

We used candidate gene-based association analyses to detect possible causal variations underlying lenticel phenotypic variation. The mixed linear model in the TASSEL ver. 5.0 software [68] was used to test the statistical associations among all common allelic variations (including SNPs and InDels) and three lenticel-related traits, as described above.

### 4.5. Gene Ontology Enrichment Analysis

All genes uncovered by GWAS were annotated on the basis of the gene models of the *P. tomentosa* genome. Gene Ontology (GO) enrichments were determined by AgriGO (http://bioinfo.cau.edu.cn/agriGO/index.php (accessed on 15 August 2019)). Significant GO annotations for a list of genes were determined by FDR threshold *p* < 0.01.

### 4.6. Epistasis Analysis of Each SNP Pair

To detect epistatic effects of locus pairs on lenticel traits, we used the EPISNP ver. 2.0 module integrated with the epiSNP software package for Windows, based on the extended Kempthorne model [43]. Only SNPs showing significant association signals (*p* ≤ 5.318 × 10^–8^) according to GWAS were used for epistasis analysis. Interactions among different loci were orthogonally decomposed into four components: additive × additive, additive × dominant, dominant × additive, and dominant × dominant interactions.

### 4.7. RNA Extraction and RNA Sequencing

We collected apexes, leaves, bark, phloem, cambium, developing xylem, mature xylem, and roots from the 5-year-old *P. tomentosa* clone ‘741’, with three biological replicates. Total RNA was extracted from shoot apical meristem (peak), leaves, bark, phloem, cambium, developing xylem, mature xylem, and roots using cetyltrimethylammonium bromide lysis buffer and the RNeasy Plant Mini kit (Qiagen, Shanghai, China) according to the manufacturer’s instructions. High-quality RNA, determined using a nanodrop spectrophotometer (Thermo Scientific, Waltham, MA, USA), and Qubit 2.0 fluorometer (Invitrogen/Life Technologies, State of California, USA) were used for cDNA library construction according to the sample preparation instructions (Illumina). Paired-end sequencing was performed using the Illumina HiSeq 4000 platform (Illumina) following the user’s manual, and 150-bp paired-end reads were generated.

### 4.8. Screening for Selective Sweeps

To identify signatures of putative genomic regions under adaptation among three climate regions of *P. tomentosa*, nucleotide diversity (π) and genetic differentiation (F_ST_) statistics were calculated using a sliding window approach (5-kb window) in the PLINK [67]. We scanned the ratios of genetic diversity between the NW and NE subpopulations (π_NW_/π_NE_), NW and S populations (π_NW_/π_S_), and NE and S populations (π_NE_/π_S_) to select windows with the top 5% ratios and the maximum absolute value of F_ST_ as candidate regions for further analysis. We performed the same subpopulation comparisons as described in Du et al. (2019) [22], because the S and NW climate regions are the central and marginal distribution regions, respectively, of *P. tomentosa* [25]. Finally, genes whose gene regions overlapped the genomic regions of adaptive selection found by GWAS were selected as candidate genes for further analysis.

### 4.9. Quantitative Reverse-Transcription Polymerase Chain Reaction (qRT-PCR)

Lenticel tissues were collected from *P. tomentosa* (accessions: 6510, 1701, 5016, 0001, 0057, 6338, 4411, 3717, and 3007) with different genotypes and were immediately frozen in liquid nitrogen. Lenticel tissue was collected by scraping thin (∼0.5 mm) slices of the lenticel surface at a plant height of 1.35 m. 6510, 6338, and 4411 came from NW climate regions; 1701, 0001, and 0057 came from NE climate regions; 5016, 3717, and 3007 came from S climate regions. Total RNA was extracted from lenticel tissue with cetyltrimethylammonium bromide lysis buffer and the RNeasy Plant Mini kit (Qiagen, Shanghai, China) according to the manufacturer’s instructions. The gene-specific primers were designed using the Primer Express ver. 5.0 software. We performed qRT-PCR using the 7500 Fast Real-Time PCR System with SYBR Premix Ex Taq, as described in Du et al. (2015) [61]. All reactions were performed in technical and biological triplicates with *Populus Actin* (EF145577) as the internal control, and the PCR program described in Zhang et al. (2011) [74].

### 4.10. PtoNAC083 and PtoMYB46 Vector Construction and Transformation in A. thaliana

Full-length cDNAs of *PtoNAC083* and *PtoMYB46* from the 1-year-old *P. tomentosa* clone ‘LM50’ were amplified and inserted into the pCXSN vector under the control of the 35S promoter and introduced into *Agrobacterium* GV3101. Wild-type (WT) *A. thaliana* (Col-0) plants were transformed using the floral dipping method. Individuals of the T3 generation were obtained using hygromycin-based selection (30 mmol/L). The expression levels of *PtoNAC083* and *PtoMYB46* in *A. thaliana* were measured in three independent homozygous transformed lines by qRT-PCR, and histological sections were used to verify the effects on the cells.

### 4.11. Paraffin Sectioning and Histological Analysis

Barks from the young stem of 1-year-old *P. tomentosa* clone ‘LM50’ were used for anatomical observation of lenticels. *A. thaliana* samples obtained from the stem at a height of 2 cm were used to verify the effects on the cells (1 cm high in stem). FAA (acetic acid; ethyl alcohol; formaldehyde) were used to fix tissue materials. For cell diameter of *A. thaliana*, twenty cells from each sample were measured and the data averaged. Hypocotyls of 13-day-old *A. thaliana* were stained with 10 μ/mL propidium iodide (PI) for 1 min and imaged using a confocal laser scanning microscope (Leica, SP8, Allendale, NJ, USA).

### 4.12. Physiological Measurements

Net CO_2_ assimilation, transpiration rate, and instantaneous leaf water-use efficiency (WUE) (Pn/Tr) of 30-day-old plants were measured by the LI-6400 portable photosynthesis system (LI-COR Inc., Lincoln, NE, USA). The measurements were taken on a fully expanded rosette leaf (six per line) at chamber temperature 23 °C and an ambient CO_2_ concentration of 360 µmol mol^−1^.

## Figures and Tables

**Figure 1 ijms-22-09249-f001:**
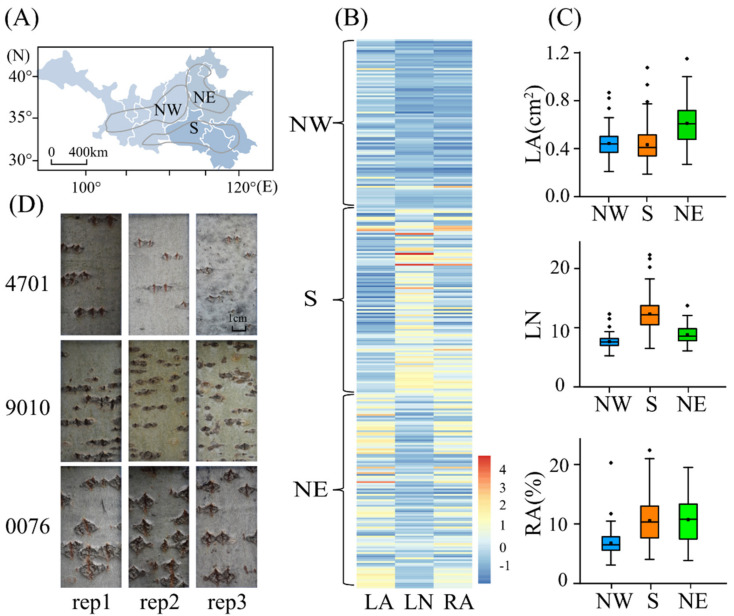
Phenotypic variation in lenticel traits in a natural population of *Populus*
*tomentosa*. (**A**) Map of climatic regionalization of the *P. tomentosa* population. Grey blocks indicate administrative divisions among provinces of China. Black lines indicate the three climate regions, based on Huang (1992). Northwest (NW), Northeast (NE), and South (S) represent the northwest, northeast, and southern climate regions, respectively. (**B**) Heatmap visualization of the association of three lenticel traits in the population. (**C**) Box plot of the three lenticel traits. (**D**) Lenticel phenotypic differences among three representative individuals (4701, 9010, and 0076) from the three climate regions. Rep1–3 represent three clones of the same genotype. Bars = 1 cm.

**Figure 2 ijms-22-09249-f002:**
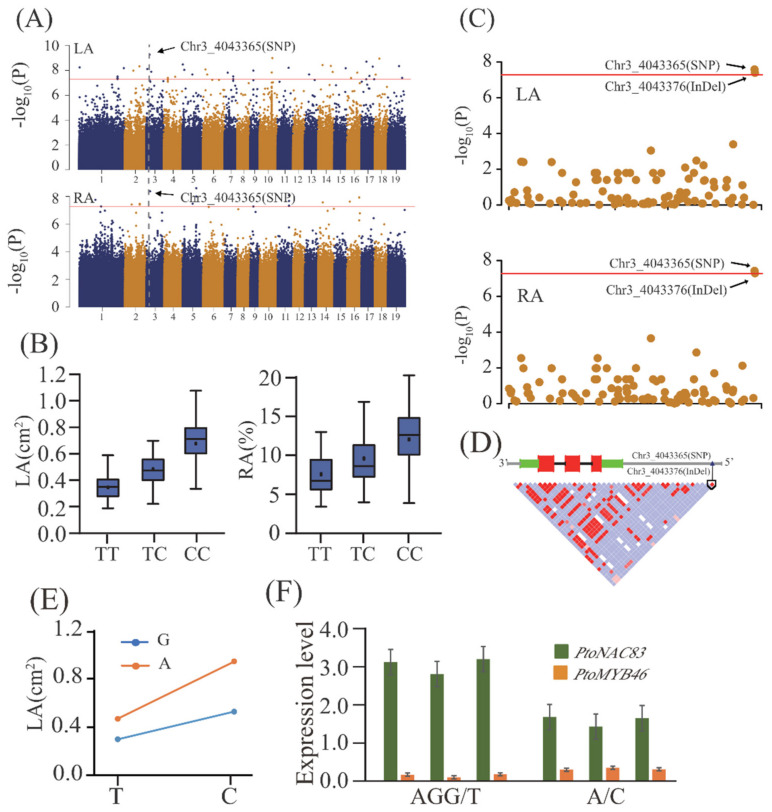
*PtoNAC83* is an important genetic regulator that contributes to lenticel area (LA) and the ratio of the total LA to the area of a standard rectangular sampling window (RA) in *Populus*
*tomentosa*. (**A**) Manhattan plots of GWAS results for all single nucleotide polymorphisms (SNPs) versus LA and RA. The horizontal red line represents the significance threshold (−log_10_P > 7.27425). The arrow indicates the signal containing the candidate genes. (**B**) Box plot of SNP genotype effects of Chr3_4043365 on LA and RA. (**C**) Manhattan plots of candidate genes associated with genetic markers (SNPs and InDels) versus LA and RA. (**D**) Gene structure diagram and linkage disequilibrium (LD) heatmap (bottom) of *PtoNAC83*. Green, red, black, and gray represent non-coding, exon, intron, and intergenic regions, respectively. (**E**) Epistatic effects between Chr3_4043365 and Chr9_3551378 underlying LA. The C allele (Chr3_4043365) depends on the A allele (Chr9_3551378) in increasing LA. **(F)** Genotype effects of haplotype on *PtoNAC83* and *PtoMYB46* expression. Data are presented as means ± SE (n = 12).

**Figure 3 ijms-22-09249-f003:**
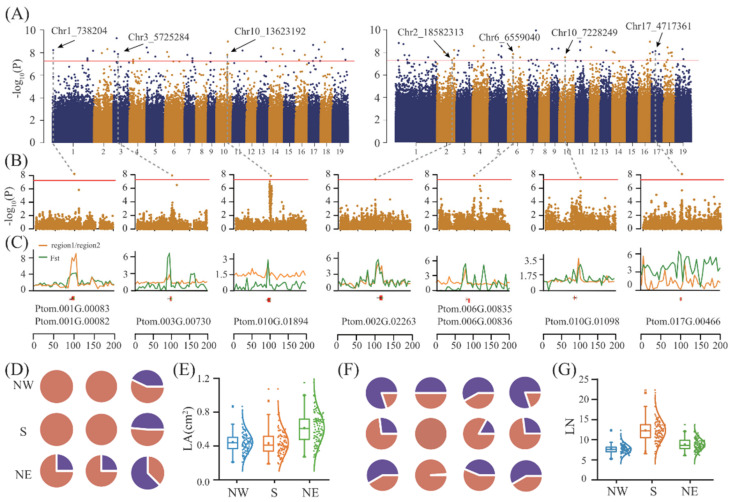
Population genetic differentiation for the 10 genes and GWAS loci with allele frequency differences across three subpopulations. (**A**) Manhattan plots of the lenticel number (LA) and lenticel number (LN). (**B**,**C**) Manhattan plots, nucleotide diversity (π), and genetic differentiation (F_ST_) values within 200 kb of the SNP loci. Yellow and green lines represent the π ratio (π_region1/π_region2) and 10× F_ST_ values, respectively. The structures of 9 genes that harbor SNPs significantly associated with lenticel traits and overlap the genomic regions of adaptive selection. Red, black, and green rectangles indicate exon, intron, and 3′UTR/5′UTR sequences, respectively. (**D**–**G**) Allele frequencies of SNPs and phenotypic variation in lenticel traits in the three subpopulations. In the pie chart, orange and purple indicate the dominant and mutant alleles, respectively.

**Figure 4 ijms-22-09249-f004:**
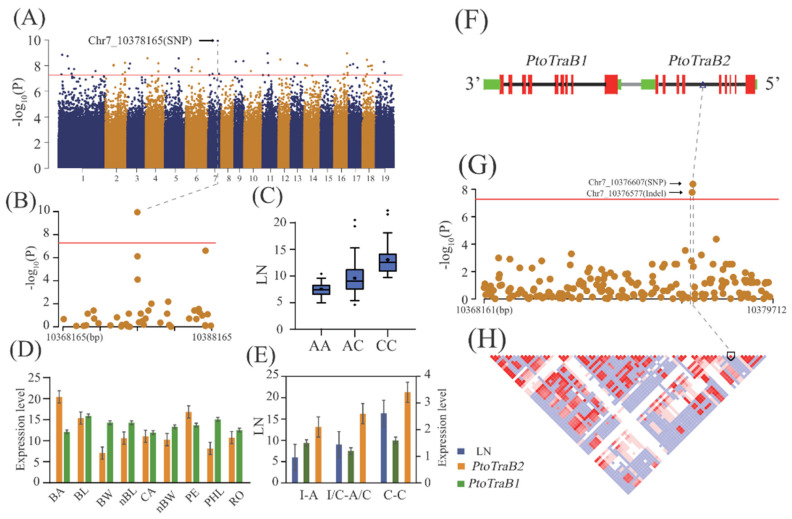
*PtoTraB2* plays an important role in the genetic regulation of LN in *Populus*
*tomentosa*. (**A**) Manhattan plots of LN. The arrow indicates the signal containing the candidate genes (Chr7_10378165). (**B**) Regional association plot of Chr7_10378165. (**C**) Box plot of genotype effects of SNP Chr7_10378165 for LN. (**D**) Expression patterns of *PtoTraB1* and *PtoTraB2* in nine tissues of *P. tomentosa.* BA, bark; BL, mature leaves; BW, mature xylem; nBL, developing leaves; CA, cambium; nBW, developing xylem; PE, phloem; PHL, phloem; RO, root. (**E**) Effects of haplotype on LN and *PtoTraB1* and *PtoTraB2* expression, where “I” represents the wild-type (WT) locus (CTTATTTCT). (**F**) Gene structure of *PtoTraB1* and *PtoTraB2*. Red, black, and green rectangles represent the exon, intron, and 3′UTR/5′UTR sequences. (**G**) Regional association plot of *PtoTraB1* and *PtoTraB2***,** including SNPs and InDels. (**H**) LD representation of pairwise r^2^ values among all polymorphic sites (SNPs and InDels) across the *PtoTraB1* and *PtoTraB2* locus, where darker red areas indicate higher r^2^ values.

**Figure 5 ijms-22-09249-f005:**
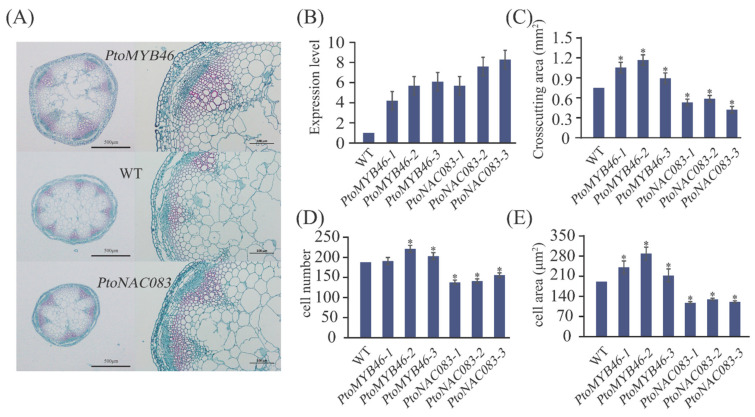
Characterization of *PtoNAC083*, *PtoMYB46*, and WT plants. (**A**) Overall morphology and stem characterization of *PtoNAC083-1*, *PtoMYB46-1*, and WT plants. (**B**) Relative expression levels of WT, *PtoNAC083* and *PtoMYB46*. (**C**–**E**) Basal stem cross-sectional area, epidermal cell number, and epidermal cell area of *PtoNAC083*, *PtoMYB46*, and WT plants. (**F**–**H**) Statistics of stomatal in in the hypocotyls. Stomatal length (**F**), Stomatal length (**G**) and Stomatal area (**H**). (**I**–**K**) Net photosynthetic rate (**I**), transpiration rate (**J**), and instantaneous leaf water-use efficiency (WUE) (**K**). *, *p* < 0.05.

**Table 1 ijms-22-09249-t001:** Full details of genes harboring significantly associated single nucleotide polymorphisms (SNPs) and overlapping the genomic regions of adaptive selection.

Phenotype	SNP	Alleles	SNP *p*-Value	Gene_Id	Regions of Selective Sweep	π_nw/π_s	Fst	Description
LA	Chr1_738204	T/C	5.76134 × 10^−9^	*Ptom.001G.00083*	Chr1: 725001–745000	6.958347576	0.40446225	
				*Ptom.001G.00082*	Chr1: 725001–745000	6.958347576	0.40446225	myb domain protein 5
LA	Chr3_5725284	G/T	1.28839 × 10^−8^	*Ptom.003G.00730*	Chr3: 5725001–5730000	2.835296259	0.662955	trehalase 1
LA	Chr10_13623192	A/C	1.49898 × 10^−8^	*Ptom.010G.01894*	Chr10: 13625001–13630000	2.504975728	0.293226	
LN	Chr2_18582313	C/G	4.69554 × 10^−8^	*Ptom.002G.02263*	Chr2: 18580001–18600000	3.568314954	0.36376425	NPL4-like protein 1
LN	Chr6_6559040	A/T	1.37452 × 10^−8^	*Ptom.006G.00835*	Chr6: 6535001–6540000	3.304231993	0.384925	Pentatricopeptide repeat (PPR) superfamily protein
				*Ptom.006G.00836*	Chr6: 6555001–6560000	3.304231993	0.384925	uracil phosphoribosyltransferase
LN	Chr10_7228249	A/T	2.67429 × 10^−8^	*Ptom.010G.01098*	Chr10: 7225001–7230000	3.587991883	0.253253	CytoChrome P450 superfamily protein
LN	Chr17_4717361	A/G	7.59074 × 10^−9^	*Ptom.017G.00466*	Chr17: 4670001–4685000	4.793217649	0.485404	P-loop containing nucleoside triphosphate hydrolases superfamily protein

π_region1/π_region2 refers to the ratio of π in different climate regions. ATG, *Arabidopsis thaliana* gene; LA, lenticel area; LN, lenticel number.

**Table 2 ijms-22-09249-t002:** Major single nucleotide polymorphisms (SNPs) linked to candidate genes significantly associated with lenticel traits in *Populus* association mapping.

Trait	SNP	Allele	Location	Marker_R^2^	SNP *p*-Value	*Populus tomentosa* Gene	ATG (*Arabidopsis thaliana* Gene)	ATG Synonyms (Abbreviation) in Plants
LA	Chr3.4043365	T/C	Promoter	0.10964	5.41161 × 10^−^^10^	*Ptom.003G.00501*	AT5G13180	NAC domain containing protein 83
LA	Chr14.5078112	C/T	Exon	0.15034	3.76948 × 10^−9^	*Ptom.014G.00612*	AT3G61870	
						*Ptom.014G.00614*		
LA	Chr2.16529660	T/G	Downstream	0.07527	4.70312 × 10^−9^	*Ptom.002G.02153*	AT1G02730	cellulose synthase-like D5
LA	Chr5.11035663	A/T	Upstream	0.09349	2.17057 × 10^−8^	*Ptom.005G.01280*	AT5G67080	mitogen-activated protein kinase kinase kinase 19
LN	Chr7.10376607	A/C	Intron	0.09408	1.15327 × 10^−10^	*Ptom.007G.00997*	AT1G05270	TraB family protein
						*Ptom.007G.00998*	AT1G05270	TraB family protein
RA	Chr5.15162313	G/A	Downstream	0.09193	2.47672 × 10^−9^	*Ptom.005G.01558*		
						*Ptom.005G.01557*	AT5G20240	K-box region and MADS-box transcription factor family protein
RA	Chr3.4043365	T/C	Downstream	0.14604	4.18145 × 10^−9^	*Ptom.003G.00501*	AT5G13180	NAC domain containing protein 83
RA	Chr14.5078112	C/T	Exon	0.13649	2.54129 × 10^−8^	*Ptom.014G.00612*	AT3G61870	
						*Ptom.014G.00614*		
RA	Chr5.11035663	A/T	Upstream	0.09769	3.17383 × 10^−8^	*Ptom.005G.01280*	AT5G67080	mitogen-activated protein kinase kinase kinase 19
RA	Chr2.16529613	C/T	Downstream	0.16756	3.22066 × 10^−8^	*Ptom.002G.02153*	AT1G02730	cellulose synthase-like D5

## Data Availability

The raw data from genome re-sequencing of P. tomentosa have been deposited in the Genome Sequence Archive (GSA) in BIG Data Center at Beijing Institute of Genomics (BIG), Chinese Academy of Sciences under the accession number of CRA000903. All data of GWAS and transcriptome are provided in Appendix A.

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
