# Peer review of "Genetic Architecture and Genome-Wide Adaptive Signatures Underlying Stem Lenticel Traits in Populus tomentosa"

_ijms, 2021, doi:10.3390/ijms22179249_

Round 1

Reviewer 1 Report

  • Authors identified 108 significant single 18 nucleotide polymorphisms within the tomentosa genome that have additive and dominant effects on the lenticel. They also identified 88 genes involved in lenticel traits. Two of these genes were cloned and overexpressed in Arabidopsis thaliana. The authors showed that these two genes have a significant effect on cell division and cell number. However, more evidence is required to support that these two genes are involved in lenticel traits.
  • Add the full description of the geographical area abbreviations when mentioned for the first time.
  • Line # 89: Add the full description of the abbreviations LA, LN, and RA
  • Figure 4 E: Add the full description of the abbreviations of the different tissues to the legend.
  • Figure 5A: What is the purpose of picture “A”? More morphological and physiological data needs to be combined with this picture.
  • How is the expression of the two identified genes at the different geographical locations and in response to various abiotic stress factors such as high and cold temperature and waterlogging/flooding?
  • The authors did not mention anything about the morphological examination of the surface of stems and roots.
  • How about the cell wall composition or any other chemicals analysis that can be used to show similarity between the natural lenticel tissue and the transgenic plants overexpressing these two genes.
  • Some homologs of the identified genes are involved in sugar metabolism; it would be better if authors provide more analysis for the transgenic plants.
  • It is true that the development involved an increase in cell number; however, the primary function of the lenticels is to act as a pathway for the direct exchange of gases between the internal tissues and atmosphere. This function helps plants to adapt under abiotic stress such as temperatures, drought, and flooding. To support the notion that these two genes are involved in lenticel traits, transgenic plants should be tested for abiotic stress tolerance.
  • Line # 553-554: Remove duplicated words “shoot apical meristem (peak), leaves, bark, phloem, cambium, developing xylem, mature xylem, and roots”

Author Response

Please see attachment file of Reply to the Review Report (Reviewer 1).

Reviewer 2 Report

This manuscript by Li et al. has interesting research results on genetic architecture and genome-wide adaptive signatures underlying stem lenticel traits in Populus tomentosa by GWAS analysis, gene expression profiling, and Arabidopsis overexpression analysis. Authors have identified the genetic architecture and adaptive mechanisms of lenticel variation. This article is well written, but it can be accepted after minor revision as follows:

Line 39: Authors need to give the full names of LA, LN, and RA as the first showing terminology.

Line 120: Authors need to describe about the GWAS analysis including the number of accessions at the first sentence or paragraph.

Line 261: It is better to begin by describing as follows: The association analysis was carried out with the Populus tomentosa population of 303 10-year-old accessions that were planted in a common garden in Guan Xian County, Shandong Province, China.

Author Response

Please see attachment file of Reply to the Review Report (Reviewer 2).

Round 2

Reviewer 1 Report

1- Thanks for providing a detailed answer for the comments, however, authors needs to integrate these answers, figures and the supported references in the manuscript to provide the readers with a complete picture.

2- Abstract needs to be rewritten to indicate the new added information.

Author Response

Please see the file of Responses to Reviewer1.
